**Data Availability Statement:** All relevant data are within the paper and its Supporting Information files.

# Low venous thromboembolism incidence in high risk medical patients in an Israeli hospital. Can risk assessment be extrapolated to different populations?

Ofir Koren[1,2], Arin Nasser[3], Mazen Elias[2,3], Gilat Avraham[3], Nahum Freidberg[1], Walid Saliba[2,4], Lee H. Goldstein[2,5]*

1 Emek Medical Center, Heart Institute, Afula, Israel, 2 Bruce Rappaport Faculty of Medicine, Technion-Israel Institute of Technology, Haifa, Israel, 3 Internal Medicine C, Emek Medical Center, Afula, Israel, 4 Translational Epidemiology Unit, Carmel Medical Center, Haifa, Israel, 5 Clinical Pharmacology Unit, Emek Medical Center, Afula, Israel

* Goldstein_le@clalit.org.il

## Abstract

### Background

Guidelines recommend venous thromboembolism (VTE) prophylaxis in hospitalized medical patients with Padua prediction score (PPS) ≥4 points. This recommendation is based on the high risk of symptomatic VTE observed among these patients in the Italian PPS derivation study, and the fivefold risk reduction with VTE-prophylaxis. This study aims to assess the incidence of VTE in high risk medical patients in a medium sized hospital in Israel.

### Method

In this retrospective cohort study, data was collected of all medical patients hospitalized between January and June 2014. Patients were classified into low and high risk groups according to their PPS score, and according to whether they received anticoagulant thromboprophylaxis for VTE. Patients were further randomly selected to compare high risk patients that did or did not receive anticoagulant thromboprophylaxis. We further compared VTE incidence in high and low risk patients not treated with thromboprophylaxis. A search was conducted for diagnoses of venous thromboembolism and death during hospitalization and the following 90 days.

### Results

568 high risk patients (PPS ≥4 points) were included, 284 treated with prophylactic anticoagulation and 284 not. There were no VTE events in either group. There was no difference in mortality. A total of 642 non anticoagulated patients were randomly selected, 474 low risk and 168 high risk. There were no VTE events in either group.

**Funding:** The author(s) received no specific funding for this work

**Competing interests:** The authors have declared that no competing interests exist.

## Conclusions

The risk of VTE appears to be very low in our study, suggesting that among medical patients with PPS ≥4, the risk of VTE may differ dramatically between populations.

## Introduction

Venous Thromboembolism (VTE) is defined as deep vein thrombosis (DVT) or pulmonary embolism or both, and is associated with increased mortality and complications such as, post-thrombotic syndrome, increased risk for recurrence of thrombosis and development of pulmonary hypertension [1].

In the past, only surgical patients were considered at risk for developing VTE [2]. Over the past few decades it has become apparent that hospitalized patients have an increased risk of developing VTE [3–8] and VTE prophylaxis confers a strong benefit in a selected high risk group of medical patients [9–15]. The recommendation for prophylaxis has been expanded to patients hospitalized in all wards, and in particular, to those hospitalized in the internal medicine wards (medical patients) [16, 17].

Various risk assessment models (RAMs) have been developed for identifying medical patients at increased risk of VTE. The Padua Prediction Score (PPS) [18], the International Medical Prevention Registry on Venous Thromboembolism (IMPROVE) [19] and the Geneva Risk model [20] are three risk assessment models (RAM) that have undergone external validation in cohorts of acutely ill hospitalized medical patients. The Padua Prediction Score incorporates 15 risk factors within 11 items, and is one of the few RAMs which have been validated in medical patients. In the Padua Prediction study VTE event rate at 90 days was 11.0% in high-risk patients (score 4 and above) without thrombo-prophylaxis as compared to 2.2% in high risk patients with thrombo-prophylaxis reflecting a relative risk reduction of 80%.

The 90 day VTE event rate in low-risk patients of whom the majority did not receive thrombo-prophylaxis was 0.3% [18]. In an additional Italian study, the results of the Padua RAM were validated and the risk of VTE in high risk medical patients reduced from 8.3% to 1.5% with appropriate anticoagulation. [21] In 2014 a multicenter validation of the Geneva Risk Score was published. When compared with the Padua Prediction Score, the Geneva score was better at identifying low-risk patients, who are not in need of thrombo-prophylaxis. An interesting finding in the Geneva Score validation study was the relatively low incidence of VTE in medical patients. In this study, which included 1478 patients hospitalized in internal medicine departments in Geneva, the VTE incidence rate, in high risk patients (PPS ≥4) was only 3.5%, whereas in the Padua Score validation studies, the VTE incidence rate was 11% [20] and 8.5% [21]. The IMPROVE RAM was derived from a registry of 15,125 medically-ill patients and includes 11 predictors. Patients at high risk for VTE are designated with a score ≥4.These patients had a VTE risk of 5.7% versus <1% in the low-risk group [19].The IMPROVEDD RAM adds D Dimer to the risk assessment model [22].

According to international guidelines, such as the guidelines of the American College of Chest Physicians (ACCP), prophylactic anticoagulation should be administered to high-risk medical patients using the Padua Score (PPS ≥4) [16]. The American Society of Hematology suggested assessing patients individual VTE risk using either the Padua or the IMPROVE or IMPPROVEDD risk RAMs taking into account the patients bleeding risk.[17] The recommended prophylaxis, among others, is a low molecular weight heparin (LMWH), such as Enoxaparin. In view of the international guidelines, calculating the Padua prediction score has

**Table 1. Padua Prediction Score.**

| Risk factor | Score |
| --- | --- |
| Active cancer and/or chemotherapy over the past six months | 3 |
| Past VTE[x] event | 3 |
| Reduced mobility over the past three days | 3 |
| Hypercoagulability | 3 |
| Trauma and/or surgery over the past month | 2 |
| Old aged (70 or more) | 1 |
| Lung and/or heart failure | 1 |
| Acute myocardial infarction and/or acute stroke | 1 |
| Acute infection and/or rheumatologic disorder | 1 |
| Obesity | 1 |
| Active hormonal treatment | 1 |

[x]VTE–Venous Thromboembolism.

been proclaimed a quality measure, by all the Clalit Health Services hospitals in Israel, our hospital included.

The objective of this study was to examine the incidence of VTE events in our medical patients stratified by the Padua score and examine the benefit of administering LMWH as prophylaxis treatment (Table 1).

## Material and methods

We conducted a retrospective observational descriptive study in Emek Medical Center, a general, 500 bed teaching hospital in the north east of Israel, belonging to the Clalit Health Services. We collected data from files of patients hospitalized in four internal medicine departments from January 1 2014 to June 30 2014 who met the study inclusion criteria (ages 18 and up with a Padua score recorded) (Table 2).

Pregnant women, patients hospitalized for deep vein thrombosis, pulmonary embolism, acute coronary syndrome and other conditions requiring therapeutic dose of anticoagulants were not included in the study. Also not included were patients with contraindications to

**Table 2. Inclusion and exclusion criteria.**

| Inclusion Criteria |
| --- |
| 1) Hospitalized in Internal Medicine departments Jan-Jun 2014 |
| 2) Age ≥18 |
| 3) Record of Padua Score |
| **Exclusion Criteria** |
| 4) Pregnant women |
| 5) Patients hospitalized for venous thromboembolism (VTE) |
| 6) Patients on anticoagulants prior to hospitalization. |
| 7) Patients discharged from hospital with an anticoagulant of any kind provided it was not prescribed for venous thromboembolism that developed over the course of the hospitalization. |
| 8) Patients who received full dose anticoagulants during hospitalization |
| 9) Patients with contraindications for anticoagulant treatment |
| a. Platelets< 50,000 |
| b. Recent acute bleeding |
| c. INR > 1.5 |

anticoagulant treatment as severe thrombocytopenia (defined as less than 50,000 platelets), INR above 1.5 and recent severe bleeding.

Thromboprophylaxis was defined as once daily treatment with an Enoxaparin, a low molecular weight heparin, at a dose of 0.5–1 mg/kg throughout the hospitalization for VTE prevention. The Padua score was calculated prospectively and entered in the electronic medical chart by the attending hospital-physicians upon patients arrival to the ward.

In order to compare VTE incidence in high risk patients, treated or not treated with thromboprophylaxis, a computerized random sampling, of 284 patients in each group was performed.

We further compared VTE incidence in high and low risk patients not treated with thromboprophylaxis by random sampling of 168 high risk patients and 474 low risk patients. The primary outcome was defined as symptomatic deep vein thrombosis or pulmonary embolism (Venous Thromboembolism–VTE) during hospitalization and over the course of a further 90 days. The secondary outcome was defined as all-cause mortality over the same time period. Since the hospital is a regional hospital which belongs to Clalit Heath Services and most of the hospitalized patients are insured by Clalit, a computerized scan of the hospital and Clalit Health Services databases was conducted in search of diagnoses of VTE, major bleeding or death. When there wasn't sufficient information regarding outcomes an experienced physician reviewed the medical files, and if need, contacted the patient. In order to detect possible cases of death due to unrecognized pulmonary embolism, cause of death was reviewed in search of unexpected death. Patients who died out of hospital, and the reason of death was unknown were also defined as unexpected death.

## Sample size

The sample size was calculated based on the incidence of VTE found in two large studies, the Italian Padua Score validation study, in which an 11% incidence rate was demonstrated, and the Swiss, Geneva Score Validation study which demonstrated an incidence of 3.2%. We estimated a VTE risk of 5% in our patients with PPS ≥4 points. In order to show that in the group that received thromboprophylaxis, the incidence of VTE decreases by approximately 80%, 568 patients were enrolled (284 in each group) in order to achieve power of 80% and alpha of 5% (two tailed test). In order to show that in the group with the low Padua score the VTE incidence is 1%, we would need to recruit 644 patients (at a 1:3 ratio, i.e. 161 patients with a high Padua score and 474 with a low Padua score), in order to achieve power 80% and alpha of 5% (two-tailed test).

## Ethics

The study was approved by Emek Medical Center Ethics Committee in accordance with the Helsinki Convention No. EMC-135-14. Informed consent was waived due to the confidentiality of patients data and the methodology of the study.

## Statistical analysis

Chi-square test was performed to analyses the association between the study groups and categorical variables. For continuous variables we used the T-test (or alternative the Wilcoxon two-sample test). Categorical variables were presented using frequencies and percent's. Continuous variables were presented using mean ± standard deviation. The statistical analyses were performed using SAS 9.4 software. P-value<0.05 was considered significant.

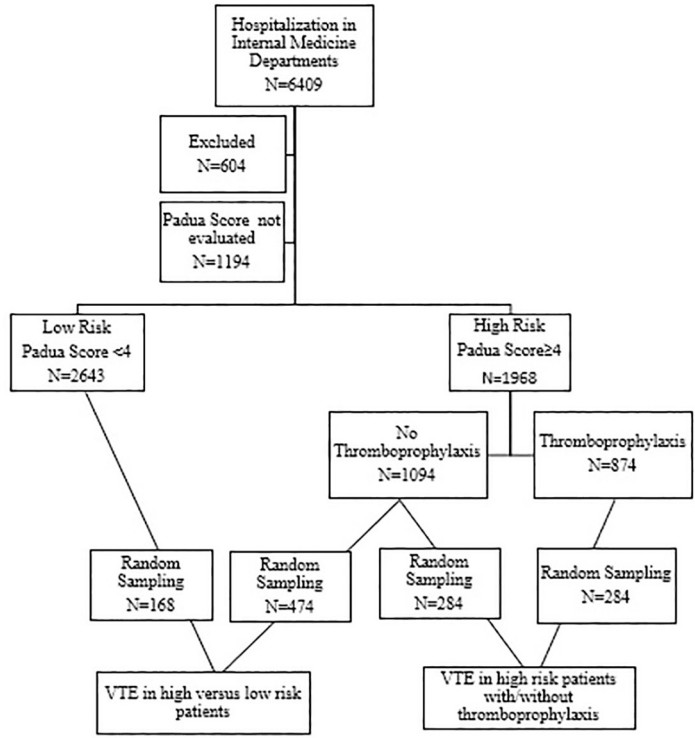

**Fig 1. Study design.**

## Results

During the first six months of 2014, there were 6409 hospitalizations in the internal medicine departments (Fig 1). Of 6409 medical patients, 4611 met the study criteria, 1968 high risk and 2643 low risk patients. Among 1968 high risk patients, 874 patients were treated with prophylactic LMWH and 1094 were not. A computerized random sampling, of 284 patients in each group was performed in order to compare VTE incidence in high risk patients, treated or not treated with thromboprophylaxis. We further compared VTE incidence in high and low risk patients not treated with thromboprophylaxis by random sampling of 474 low risk patients and 168 high risk patients.

For the high risk group comparison between thromboprophylaxis /non thromboprophylaxis the average age in both groups was 75 years of age. There was no difference in the basic characteristics of the two groups, including: gender, age, weight and primary admission diagnosis (Table 3).

The average Padua score was higher in patients treated with LMWH as opposed to non-treated group (5.77±1.68, 5.15±1.29 [P<0.001]) respectively. No significant correlation was detected between the Padua score and duration of hospitalization (P = 0.1191, with a correlation coefficient of 0.06547). The hospital stay in the treated group was longer than the stay in the non-treated group (6.82±5.6, 4.9±3.8 days respectively [P = 0.0009]). No VTE events were detected during hospitalization and over the course of 90 days in both groups (Table 4).

Ninety-day all-cause mortality rate was 65/284 (22.9%) in the group treated with prophylactic LMWH, compared to 53/284 (18.7%) in the non-treated group (P = 0.21). No significant difference was detected in the incidence of unexpected deaths between groups. (10/284 (3.52%) in the group that received thromboprophylaxis and 7/284 (2.46%) in the non-treated group. (p = 0.46)

**Table 3. Patients characteristics.**

| Patients Characteristics | | High Risk Patients with/without thromboprophylaxis | | High versus low risk patients (without thromboprophylaxis) | |
|---|---|---|---|---|---|
| | | Prophylactic LMWH N = 284 | No Prophylactic LMWH N = 284 | Padua score≥4 N = 168 | Padua score<4 N = 474 |
| Age **(average (SD)** | | 75.5(13.3) | 74.2(13.6) | 75.3± 13.6 | 61.1± 17.5 |
| Gender (male) N **(%)** | | 128 (45.10) | (140) 49.60 | 85 (50.6) | 281 (59.3) |
| Weight (kg) **(average (SD))** | | 77.2±17.7 | 74.5±17.3 | 74.3± 17.8 | 78.2± 17.9 |
| Duration of hospitalization (days) **(average (SD))** | | 6.82±5.6 | 4.9±3.8 | 5.3 ± 4.8 | 3.9± 3.2 |
| Padua Score **(average (SD), [Median (min,max)])** | | 5.77(1.68) [5(4,14)] | 5.15(1.29) [5(4,11)] | 5.4(1.56) [5 (4,11)] | 1.14(1.04) [1(0,3)] |
| **Primary Diagnosis N (%)** | Infection/sepsis [X] | 111 (39) | 102 (36) | 24(14) | 48 (10) |
| | Cardiovascular^ | 40(14) | 39 (14) | 32 (19) | 136(29) |
| | Neurological* | 37(13) | 31(11) | 24 (14) | 79 (17) |
| | Pulmonary disease[¥] | 31 (11) | 29 (10) | 44 (26) | 78(16) |
| | Hemato-oncological disease | 11 (4) | 20 (7) | 9 (5) | 10 (2) |
| | Acute renal failure [Ψ] | 3(1) | 3 (1) | 6 (4) | 15 (3) |
| | Acute rheumatologic disorder | 3 (1) | 1 (0) | 1 (1) | 12 (3) |
| | Other | 48 (17) | 59 (21) | 28(17) | 96 (19) |

Average ± standard deviation.

[X] Infectious and sepsis; pneumonia, urinary tract infection, cellulitis.

^Cardiovascular- congestive heart failure, ischemic heart disease, tachy/bradyarrhythmias.

*Neurological and cerebrovascular accident; Transient ischemic attack, syncope, vertigo.

[Ψ]–Calculated with the Cockroft-Gault formula.

[¥]Pulmonary- asthma, COPD, acute bronchitis, upper respiratory tract infection.

As expected, for the comparison between high and low risk patients (all of whom were not treated by, thromboprophylaxis), the two groups differed in most indices (Table 3). The high risk patients were on average older (75.3±13.6years) compared to (61.1±17.5years) (p<0.001), and with lower weight (74.3±17.8kg) as opposed to (78.2±17.9 kg) (p = 0.01). The groups also differed in terms of primary admission diagnosis and length of hospital stay. The average hospitalization length was 5.3±4.8 days in the high-risk patients compared to 3.9±3.2 in the low risk patients (p = 0.0005). No VTE events occurred in either high or low risk patients during hospitalization or the 90-day post hospitalization follow-up (Table 4). The mortality rate was, as expected, higher in the high risk group 35/168 (20.4%) in comparison to the low risk group 17/474 (3.6%), p<0.001. The odds of mortality was 5.6 folds higher in high risk patients compared to low risk patients (95% CI: 3.1–10.3, P = 0.001). The results were similar after

**Table 4. VTE and death within 90 days.**

| | High Risk Patients with/without thromboprophylaxis | | | High versus low risk patients (without thromboprophylaxis) | | |
|---|---|---|---|---|---|---|
| | Prophylactic LMWH N = 284 | No Prophylactic LMWH N = 284 | P Value | Padua score≥4 N = 168 | Padua score<4 N = 474 | P Value |
| **VTE** | 0 | 0 | | 0 | 0 | |
| **Death within 90 days** | 65 (22.9%) | 53 (18.7%) | 0.21 | 35 (20.0%) | 17(3.6%) | <0.001 |

controlling for hemoglobin, length of hospitalization and eGFR; OR = 3.05 (95%CI: 1.58–5.9 (P = 0.001).

No difference was observed in the incidence of unexpected deaths in the low risk in comparison to the high risk group (7/474 (1.46%) and 6/168 (3.57%) respectively (p = 0.11). Likewise, no difference in the incidence of unexpected death was observed in the high risk patients with or without thromboprophylaxis. (10/284 (3.52%) and 7/284 (2.46%) respectively, p = 0.62)

## Discussion

No VTE events were detected in our cohort of medical patients, despite some being defined as high risk patients according to the Padua risk score (PPS ≥4) and not treated by thromboprophylaxis. In other words, a Padua score greater than or less than four was not able to predict who would suffer of VTE in our patients who didn't receive prophylactic anticoagulant treatment.

Our findings suggest a low risk of VTE in our medical patients with PPS≥4 in contrary to the findings of previous studies that showed a high risk of VTE (11%). We additionally found no difference between low and high risk patients, not treated with anticoagulation, nor between high risk patients treated/non treated with thromboprophylaxis.

The 90-day mortality rate in the high-risk patients was twice higher than the low risk patients in this cohort. The Padua score is based on medical history and age, so it stands to reason that the higher the score, the higher the mortality rate. Exploring those cases reveal 8 unexpected death in the high-risk group (8/35, 22.8%) and 4 unexpected deaths in the low-risk group (4/17, 22.5%). This indicate no difference between the high and low risk groups in the incidence of unexpected death.

The discrepancy between the results of our study to those of the original Padua study could be explained if the incidence of VTE in our population of medical patients is lower than the rate described in previous studies. In retrospect, it is possible that a sample size calculated on the basis of a VTE incidence rate of 5% was insufficient, and that the rate of VTE in our population is much lower, and our sample size was not large enough to detect such a small difference. The Italian population of Padua study possibly have a higher thrombosis risk, as is demonstrated in the Geneva study [20], which attempted to validate a newly proposed risk assessment score (the Geneva Score), comparing the Padua and Geneva scores in their study.

The incidence of VTE in high risk patients in the Geneva population was considerably lower than the Padua population (3% versus 11%). These two trials were performed in specific populations (Italian and Swiss). The incidence of symptomatic VTE was considerably lower in major multi-national and multicenter studies, 0.7% in high risk medical patients in the MEDE-NOX study [23], the landmark trial for the prevention of VTE in medical patients, 0.9% in the PREVENT study [24], and 3% in the ARTEMIS study [25].

A possibly higher prevalence of VTE in the Italian population is further supported by the results of the PESIT Trial, from Padua Italy [26], which demonstrated an unexpectedly high incidence of pulmonary embolism in patients hospitalized for syncope. The incidence of pulmonary embolism was 17.3% in this Italian cohort, previously reported as 1.6% in an Iranian cohort [27], 2.5% in a Belgian cohort [28] and 1% in a Swiss cohort [29].

The lack of VTE cases in our study suggests that the incidence of VTE in our population is even lower than that observed in the Geneva study and therefore it is unclear whether using this score would be helpful to successfully predict the incidence of VTE in our patient population.

Another possible explanation for the difference in VTE incidence could be if the risk assessment which was performed by the attending physician was done inaccurately regarding

mobility assessment. Of all the 11 variables in the Padua Score, the only subjective variable, which could possibly be evaluated incorrectly, is the mobility score. This variable contributes 3 points to the score if the patient incapable to walk any further than the toilet and back, and not, as might be erroneously evaluated as a totally bedridden patient. We assume that if the immobility score was performed inaccurately, the patient would have been evaluated with a lower and not higher score. Thus, our high risk patients would have been of a very high risk, making our results even more significant.

## Limitations

This was a retrospective study which on one hand presents a disadvantage in design but on the other hand, works to our advantage since there was no need to obtain the patients signed consent to participate in the study as required in the previous studies (Padua and Geneva). Hence, our study included seriously ill patients at high risk of symptomatic VTE and VTE related death in which we would not have been able to obtain signed consent if a prospective design was adopted. This population probably reflects reality more accurately.

Data was collected from a computerized system in a retrospective manner which may underestimate the true incidence of VTE. Clalit Health Services is the largest of the four health funds with around 3.8 million insured members, 54% of the Israeli population. The HMO's computer system constantly updating medical information from all medical services including other health service organizations and therefore the risk of missing vital information is significantly low. Furthermore, when we suspected that vital medical information was incomplete or missing, we contacted the patient directly.

In retrospect, the number of recruits was too low to prove a difference between the groups due to the low VTE incidence revealed in our study, nonetheless, it was high enough to show a very low incidence of VTE, probably even less than 0.5%. Another limitation is that the population sampled was the population for whom a Padua score had been determined. The Padua score is currently implemented on approximately 95% of the patients hospitalized at our center. During the study period (2014), implementation was lower and Padua score was not determined for 1194 patients. It is possible that this fact had an effect on the random sample.

In summary, there was no difference in the incidence of symptomatic VTE in high risk medical patients with a Padua score of 4 and above treated with prophylactic LMWH in comparison to high risk medical patients who were not treated, and there was no difference between high and low risk patients in VTE incidence. The VTE incidence rate was very low. The findings of this study imply that the risk benefit ratio of prophylactic treatment with LMWH based on the Padua score in our patients should be reassessed.

## Supporting information

**S1 Data.**
(XLS)

## Author Contributions

**Conceptualization:** Ofir Koren, Mazen Elias, Walid Saliba, Lee H. Goldstein.

**Data curation:** Ofir Koren, Arin Nasser, Nahum Freidberg, Lee H. Goldstein.

**Formal analysis:** Ofir Koren, Arin Nasser, Gilat Avraham, Nahum Freidberg, Lee H. Goldstein.

**Funding acquisition:** Ofir Koren.

**Investigation:** Ofir Koren.

**Methodology:** Ofir Koren, Mazen Elias, Gilat Avraham, Walid Saliba, Lee H. Goldstein.

**Resources:** Ofir Koren, Arin Nasser.

**Supervision:** Mazen Elias.

**Validation:** Ofir Koren.

**Writing – original draft:** Ofir Koren, Gilat Avraham, Nahum Freidberg, Walid Saliba, Lee H. Goldstein.

**Writing – review & editing:** Ofir Koren, Mazen Elias, Gilat Avraham, Nahum Freidberg, Walid Saliba, Lee H. Goldstein.

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
