## [Decision Letter · Decision Letter 0]

26 May 2020

PONE-D-20-10275

Low Venous Thromboembolism Incidence in High Risk Medical Patients in an Israeli Hospital. Can Risk assessment be extrapolated to different populations?

PLOS ONE

Dear Dr. Koren,

Thank you for submitting your manuscript to PLOS ONE. After careful consideration, we feel that it has merit but does not fully meet PLOS ONE’s publication criteria as it currently stands. Therefore, we invite you to submit a revised version of the manuscript that addresses the points raised during the review process.

The reviewers raised some concerns that should be addressed before we can reconsider your manuscript.

We look forward to receiving your revised manuscript.

Kind regards,

Prof. Raffaele Serra, M.D., Ph.D

Academic Editor

PLOS ONE

Additional Editor Comments:

the article is potentially interesting for the journal, but it requires some extra work in order to be reconsidered.

Journal Requirements:

2. Thank you for your ethics statement:

"The study was approved by the Ethics Committee of the hospital in accordance with the Helsinki Convention No. EMC-135-14. Informed consent was waived due to the confidentiality of patients data and the methodology of the study."

Once you have amended this statement in the Methods section of the manuscript, please add the same text to the “Ethics Statement” field of the submission form (via “Edit Submission”).

*For additional information about PLOS ONE ethical requirements for human subjects research, please refer to **http://journals.plos.org/plosone/s/submission-guidelines#loc-human-subjects-research**.*

3. Please include a copy of Table 2 which you refer to in your text on page 13 and 25.

Reviewers' comments:

Reviewer's Responses to Questions

**Comments to the Author**

1. Is the manuscript technically sound, and do the data support the conclusions?

Reviewer #1: Partly

Reviewer #2: Yes

2. Has the statistical analysis been performed appropriately and rigorously? 

Reviewer #1: Yes

Reviewer #2: Yes

3. Have the authors made all data underlying the findings in their manuscript fully available?

Reviewer #1: No

Reviewer #2: Yes

4. Is the manuscript presented in an intelligible fashion and written in standard English?

Reviewer #1: Yes

Reviewer #2: Yes

5. Review Comments to the Author

Reviewer #1: O Koren et al explore in a retrospective cohort study (1) whether the risk of hospital-associated VTE is 80% lower among inpatients with a high Padua score with thromboprophylaxis, compared with those without thromboprophylaxis, and (2) what the comparative VTE risk is in those with a low Padua score and a high Padua score, without thromboprophylaxis. The design is interesting and sound, with random samplings from a 6-month period in 2014 in a teaching hospital in Israel. Given that no VTE case was identified, these planned analyses become somewhat unrelevant. Nevertheless, this challenges the importance of hospital-associated VTE, and therefore indirectly of thromboprophylaxis. These provocative findings require a very strong methodology, and several major comments should be addressed.

Major comments

1. Identification and capture of VTE events

The main limitation of this study is its retrospective design. It is very surprising that there were no in-hospital diagnoses of pulmonary embolism among >700 medical inpatients. Reporting the number of chest CT and leg compression ultrasounds performed in this sample would be very important, given their routine use for in-hospital care.

Also, the capture of VTE events after discharge is not convincing enough. Authors should provide more details on the Clalit Health Services database, and how confident they can be not to have missed VTE events in the retrospective follow-up. Exploring “unexpected” death remains quite vague, especially retrospectively, as fatal PE can occur in patients with co-morbidities and not be recognized.

2. Representativeness of medical inpatients

The authors could provide more characteristics of the participants, such as the individual components of the Padua score. Some characteristics appear different from the Padua or ESTIMATE cohorts the authors cite, such as the low proportion of cancer (4-7%, vs. 20-27%). Overall, I would like to be convinced that included participants were acutely ill medical inpatients, and not rehabilitation or more geriatric inpatients. A sampling-weighted mortality estimated would also be interesting here.

3. Definition of thromboprophylaxis

Could the authors provide the study operational definition for thromboprophylaxis? Was there a minimum duration, was it for specific drugs or doses?

4. Calculation of the Padua score

The authors provide little information on how the score was calculated. Given that its evaluation is part of the inclusion criteria, was it prospectively calculated and entered in the electronic medical chart by in-charge hospital-physicians? If it was retrospectively calculated, there is a potential for misclassification that goes beyond of the immobility item and this should be discussed. What definition of immobility did the authors use?

5. Choice of the Padua score

Why was the Padua score selected for this analysis? Other scores are valid and more simple to implement, such as the Improve score (Spyropoulos, Chest 2011) or the simplified Geneva score (Blondon, JTH 2020).

Minor comments

Did none of the low-risk Padua score inpatients receive thromboprophylaxis?

Was extended thromboprophylaxis used in some participants?

Why did the authors choose a time period in 2014, for a 2020 study?

I do not understand the second sample size calculation “In order to show that in the group with the low Padua score the VTE incidence is 1%, we would

need to recruit 644 patients (at a 1:3 ratio, i.e. 161 patients with a high Padua score and 474 witha low Padua score), in order to achieve power 80% and alpha of 5% (two-tailed test).” Did the author forget to describe the comparison arm here?

Were the 474 and 284 randomly sampled high-risk Padua score without thromboprophylaxis independent, or are there same individuals in both groups?

I don’t understand why the assessment of mobility by the attending physician would change the observed incidence of VTE, given that most participants did not receive thromboprophylaxis anyway.

Table 1: past VTE instead of past deep VTE

Table 2 is improperly labeled as table 1.

Reviewer #2: Great manuscript in general;Fits into the difference the other manuscript the author mention (Geneva Score)

One comment the authors should address: Could a difference with regard to the underlying disease being responsible for the difference within the studies; eg in this analysis there is a really small number of renal disease patients, hematooncological patients.

6. PLOS authors have the option to publish the peer review history of their article (what does this mean?). If published, this will include your full peer review and any attached files.

Reviewer #1: No

Reviewer #2: No

---

## [Author Response · Author response to Decision Letter 0]

2 Jun 2020

Monday, June 1, 2020

Dear PLOS ONE Editorial Office, 

First, I would like to thank the reviewers for their comprehensive and meticulous review and the PLOS ONE journal editors for their decision to reconsider the article for publication.

General comments: 

1. The manuscript underwent extensive changes based on reviewer's comments. All changes have been highlighted in the corresponding file and labeled "revised manuscript with track changes". I will address to the specific points of the reviewers in the following paragraph. Additional unmarked version of the article will also be submitted. 

2. The financial disclosure was not change 

3. The manuscript has been formatted according PLOS ONE style. Tables were incorporated into text; figures were removed within the manuscript file and formatted according PACE digital diagnostics in separate files. Reference has been formatted. 

4. Repository Data will be available upon acceptation. I do not have a DOI or website to upload the repository data. I will need your assistance with that. 

5. The full name of the ethics committee was included in the Methods section of the manuscript and in the submission form. 

6. The Authors’ affiliations were corrected to ensure that each author is linked to an affiliation. 

I will address each point of the reviewers. 

Reviewer #1: 

Major comments

1. Identification and capture of VTE events - The main limitation of this study is its retrospective design. It is very surprising that there were no in-hospital diagnoses of pulmonary embolism among >700 medical inpatients. Reporting the number of chest CT and leg compression ultrasounds performed in this sample would be very important, given their routine use for in-hospital care. Also, the capture of VTE events after discharge is not convincing enough. Authors should provide more details on the Clalit Health Services database, and how confident they can be not to have missed VTE events in the retrospective follow-up. Exploring “unexpected” death remains quite vague, especially retrospectively, as fatal PE can occur in patients with co-morbidities and not be recognized. – 

a. “The main limitation of this study is its retrospective design” – We address this in the limitation paragraph. “This was a retrospective study which on one hand presents a disadvantage in design but on the other hand, works to our advantage since there was no need to obtain the patients signed consent to participate in the study as required in the previous studies (Padua and Geneva). Hence, our study included seriously ill patients at high risk of symptomatic VTE and VTE related death in which we would not have been able to obtain signed consent if a prospective design was adopted. This population probably reflects reality more accurately”. 

b. “It is very surprising that there were no in-hospital diagnoses of pulmonary embolism among >700 medical inpatients. Reporting the number of chest CT and leg compression ultrasounds performed in this sample would be very important, given their routine use for in-hospital care” – Our study, as opposed to others, didn’t search for asymptomatic VTE. The routine use of VTE diagnosis used in our hospital are Chest CTA for PE and US doppler for Distal DVT when suspected. Alternative modalities such as Transesophageal Echocardiography and Ventilation perfusion scan are used when C/I appear as per guidelines. We cannot provide information regarding patients who seek medical care or complain due to PE or DVT and discharge with a wrong diagnosis. It is unlikely that patients will not seek medical therapy again when PE/DVT was missed. 

c. “The capture of VTE events after discharge is not convincing enough. Authors should provide more details on the Clalit Health Services database, and how confident they can be not to have missed VTE events in the retrospective follow-up” We address this issue in the limitation section and updated the paragraph. “Clalit Health Services is the largest of the four health funds with around 3.8 million insured members, 54% of the Israeli population. The HMO's computer system constantly updating medical information from all medical services including other health service organizations and therefore the risk of missing vital information is significantly low. Furthermore, when we suspected that vital medical information was incomplete or missing, we contacted the patient directly”.

d. Exploring “unexpected” death remains quite vague, especially retrospectively, as fatal PE can occur in patients with co-morbidities and not be recognized. – All fatality cases were explored reveal 8 unexpected death in the high-risk group (8/35, 22.8%) and 4 unexpected deaths in the low-risk group (4/17, 22.5%). This indicates no difference between the high and low risk groups in the incidence of unexpected death. The paragraph has been changed accordingly. 

2. Representativeness of medical inpatients - The authors could provide more characteristics of the participants, such as the individual components of the Padua score. Some characteristics appear different from the Padua or ESTIMATE cohorts the authors cite, such as the low proportion of cancer (4-7%, vs. 20-27%). Overall, I would like to be convinced that included participants were acutely ill medical inpatients, and not rehabilitation or more geriatric inpatients. A sampling-weighted mortality estimated would also be interesting here. – Data obtained from four internal medicine departments. Geriatric patients were included since they are hospitalizing in internal medicine wards in our medical center. Rehabilitation patients were not included, they hospitalize in different ward. We do not have oncology ward so patients with VTE due to malignancy are hospitalizing in internal medicine wards. The average age was 75y. The proportion of cancer patients in our study, in our opinion, reflect the reality better than cited reports. 

3. Definition of thromboprophylaxis - Could the authors provide the study operational definition for thromboprophylaxis? Was there a minimum duration, was it for specific drugs or doses? Thromboprophylaxis was defined as once daily treatment with an Enoxaparin, a low molecular weight heparin, at a dose of 0.5-1 mg/kg throughout the hospitalization for VTE prevention. This sentence was added to the method paragraph. 

4. Calculation of the Padua score - The authors provide little information on how the score was calculated. Given that its evaluation is part of the inclusion criteria, was it prospectively calculated and entered in the electronic medical chart by in-charge hospital-physicians? If it was retrospectively calculated, there is a potential for misclassification that goes beyond of the immobility item and this should be discussed. What definition of immobility did the authors use? The Padua score was calculated prospectively and entered in the electronic medical chart by the attending hospital-physicians upon patients arrival to the ward. This sentence was added to the method paragraph. 

5. Choice of the Padua score - Why was the Padua score selected for this analysis? Other scores are valid and more simple to implement, such as the Improve score (Spyropoulos, Chest 2011) or the simplified Geneva score (Blondon, JTH 2020). – We agree that all predictive models should be re-asses geographically for cost-benefit, efficiency and safety. We analyzed the Padua model since clalit health service, as with other HMO worldwide, proclaimed this model as quality measure for internal wards

Minor comments 

1. Did none of the low-risk Padua score inpatients receive thromboprophylaxis? – Correct 

2. Was extended thromboprophylaxis used in some participants? – Thromboprophylaxis defined as VTE treatment throughout the hospitalization and not further. 

3. Why did the authors choose a time period in 2014, for a 2020 study? – It was the second year that all of the medical records were computerized, and the Padua score was applied. 

4. I do not understand the second sample size calculation “In order to show that in the group with the low Padua score the VTE incidence is 1%, we would need to recruit 644 patients (at a 1:3 ratio, i.e. 161 patients with a high Padua score and 474 with a low Padua score), in order to achieve power 80% and alpha of 5% (two-tailed test).” Did the author forget to describe the comparison arm here? Comparison arm was done in the first sentences of the paragraph comparing the PEP in treated vs non-treated groups. “We estimated a VTE risk of 5% in our patients with PPS ≥4 points. In order to show that in the group that received thromboprophylaxis, the incidence of VTE decreases by approximately 80%, 568 patients were enrolled (284 in each group) in order to achieve power of 80% and alpha of 5% (two tailed test)”. The second analyses were comparing the High vs Low Risk group. 

5. Were the 474 and 284 randomly sampled high-risk Padua score without thromboprophylaxis independent, or are there same individuals in both groups? Same individuals in both groups

6. I don’t understand why the assessment of mobility by the attending physician would change the observed incidence of VTE, given that most participants did not receive thromboprophylaxis anyway. – Mobility assessment would contribute 3 points to the Padua score. Incorrect filling of this criteria will result in placing the patient to the wrong risk group which then influence whether the patient will receive or not receive thromboprophylaxis. This may affect the incidence of the primary end points. Therefore, we explained it in the limitation paragraph. 

7. Table 1: past VTE instead of past deep VTE – Corrected 

8. Table 2 is improperly labeled as table 1. – Corrected 

Reviewer #2: 

1. Could a difference with regard to the underlying disease being responsible for the difference within the studies; eg in this analysis there is a really small number of renal disease patients, hematooncological patients. – The data in the study were randomly taken from four internal wards and included geriatric patients. The prevalence of co-morbidities among our patients did not differ when we compared it to other periods (we asses it follow reviewer comment) and we believe it faithfully represent our real population. Our patients may differ from other population worldwide, but this is exactly the message we want to convey from our research. Prior implement comprehensive and routinely treatment strategy for all hospitalized patients, the real incidence of VTE should be re-asses based on specific population or geographical areas. Previous studies have demonstrated a marked difference in the incidence of VTE events, some of which are probably explained by differences in the patient population. 

Sincerely, 

Dr. Koren Ofir

---

## [Decision Letter · Decision Letter 1]

22 Jun 2020

Low Venous Thromboembolism Incidence in High Risk Medical Patients in an Israeli Hospital. Can Risk assessment be extrapolated to different populations?

PONE-D-20-10275R1

Dear Dr. Koren,

We’re pleased to inform you that your manuscript has been judged scientifically suitable for publication and will be formally accepted for publication once it meets all outstanding technical requirements.

Kind regards,

Raffaele Serra, M.D., Ph.D

Academic Editor

PLOS ONE

Additional Editor Comments (optional):

amended manuscript is acceptable.

Reviewers' comments:

Reviewer's Responses to Questions

**Comments to the Author**

1. If the authors have adequately addressed your comments raised in a previous round of review and you feel that this manuscript is now acceptable for publication, you may indicate that here to bypass the “Comments to the Author” section, enter your conflict of interest statement in the “Confidential to Editor” section, and submit your "Accept" recommendation.

Reviewer #2: All comments have been addressed

2. Is the manuscript technically sound, and do the data support the conclusions?

Reviewer #2: Yes

3. Has the statistical analysis been performed appropriately and rigorously? 

Reviewer #2: Yes

4. Have the authors made all data underlying the findings in their manuscript fully available?

Reviewer #2: Yes

5. Is the manuscript presented in an intelligible fashion and written in standard English?

Reviewer #2: Yes

6. Review Comments to the Author

Reviewer #2: no further comments, all comments addresses in adequate manner and therefore the manuscript should be accepted as it is now

7. PLOS authors have the option to publish the peer review history of their article (what does this mean?). If published, this will include your full peer review and any attached files.

Reviewer #2: No

---

## [Editor Report · Acceptance letter]

23 Jun 2020

PONE-D-20-10275R1 

Low Venous Thromboembolism Incidence in High Risk Medical Patients in an Israeli Hospital. Can Risk assessment be extrapolated to different populations? 

Dear Dr. Koren:

I'm pleased to inform you that your manuscript has been deemed suitable for publication in PLOS ONE. Congratulations! Your manuscript is now with our production department. 

Kind regards, 

on behalf of

Prof. Raffaele Serra 

Academic Editor

PLOS ONE